# Determining the Antibiofilm Efficacy of Oregano Gel in an Ex Vivo Model of Percutaneous Osseointegrated Implants

**DOI:** 10.3390/microorganisms10112133

**Published:** 2022-10-28

**Authors:** Jemi Ong, Rose Godfrey, Brad Isaacson, Paul Pasquina, Dustin Williams

**Affiliations:** 1Department of Biomedical Engineering, University of Utah, Salt Lake City, UT 84112, USA; 2Department of Orthopaedics, University of Utah, Salt Lake City, UT 84112, USA; 3The Geneva Foundation, Tacoma, WA 98402, USA; 4Department of Physical Medicine and Rehabilitation, The Musculoskeletal Injury Rehabilitation Research for Operational Readiness (MIRROR), Uniformed Services University, Bethesda, MD 20814, USA; 5The Center for Rehabilitation Sciences Research, Uniformed Services University, Bethesda, MD 20814, USA; 6Department of Rehabilitation, Walter Reed National Military Medical Center, Bethesda, MD 20814, USA; 7Department of Pathology, University of Utah, Salt Lake City, UT 84112, USA

**Keywords:** osseointegration, biofilms, oregano, infection

## Abstract

Biofilm contamination is common in patients with percutaneous osseointegrated (OI) implants, leading to frequent infections, irritation, and discomfort. Reported infection rates soar up to 65% as the recalcitrant nature of biofilms complicates treatment. There is persistent need for therapies to manage biofilm burden. In response, we formulated and tested oregano essential oil in a topical gel as a potential biofilm management therapy. We developed an ex vivo system based on an established ovine OI implant model with *Staphylococcus aureus* ATCC 6538 biofilms as initial inocula. Gel was administered to the samples across a period of five days. Samples were quantified and colony forming unit (CFU) counts were compared against a positive control (initial bacterial inocula without treatment). Significant biofilm reduction was observed in samples treated with oregano gel compared to controls, demonstrating the potential of an oregano oil-based gel as a biofilm management therapy at the skin-implant interface of percutaneous OI implants.

## 1. Introduction

Percutaneous devices are highly susceptible to biofilm-related infections, resulting in persistent medical problems affecting up to 60–70% of patients [1,2,3]. This is attributed to a breach in the skin barrier, resulting in a constant risk of contamination by microbial flora naturally dwelling in hair follicles and sebaceous glands, or bacteria encountered from the environment during normal daily activities [4]. Transfemoral osseointegrated (OI) implants are becoming increasingly common, yet remain challenged by infection due to the percutaneous nature of the technology, which provides a prosthetic docking site allowing direct mechanical loading of the skeleton. Thus, while offering improved functionality in comparison to traditional prosthetic socket systems, management of biofilm formation and subsequent infection at the skin-implant interface of OI implants remains a major hurdle to broader adoption and patient use.

Infection rates in transfemoral OI implant systems are reported to be as high as 36% [5,6]. These prevalent and difficult-to-treat infections can be attributed to biofilms, which are well-organized communities of microbes encased in a matrix of extracellular polymeric substances (EPS), and display stark differences in comparison to free-floating bacterial cells (planktonic bacteria) [7]. Subsets of cells within a biofilm reside in a lower metabolic state and cells deep within the community are oxygen limited [8]. These characteristics contribute to the recalcitrant nature of biofilms, leading to infection relapse despite antibiotic treatment [9,10]. Furthermore, biofilms preferentially adhere to solid surfaces [11], making titanium (Ti) OI implants an ideal nidus for colonization. As a result, designing methodologies to manage biofilms at the skin-implant interface of OI implants is a major point of interest.

Considering the growing issue of antibiotic resistance, we sought to develop alternative options to manage biofilm burden. In this study, we assessed oregano essential oil formulated in a sodium hyaluronate (HA)-based gel. Essential oils are typically defined as volatile oils with strong aromatic components and are the byproducts of plant metabolism, often exhibiting natural antimicrobial properties [12,13]. Oregano oil in particular has been shown to draw its main antimicrobial potential from its constituent, carvacrol. This compound has demonstrated strong antioxidant and antimicrobial properties, with both hydrophobic and hydrophilic attributes, and reduced risk of resistance [14,15,16].

Furthermore, oregano essential oil is readily available and is often recommended for use in “home remedies” against mild infection, including those experienced by OI implant patients. As a home remedy, concentration recommendations are either vague or can range all the way up to 50% [17,18,19,20,21]. In contrast, scientific studies utilizing oregano have primarily focused on concentrations between 0.001 and 1% against planktonic bacteria and up to 12.5% in a wound healing experiment with basic antimicrobial aspects [22,23,24,25]. We therefore sought to determine if the concentrations tested in the literature are sufficient to manage established biofilms at the skin-implant interface of OI implants or if higher concentrations observed in home remedies are necessary.

We tested oregano oil formulated as the active agent of the topical gel in both single- and multi-day experiments against *Staphyloccocus aureus* ATCC 6538 biofilms in an ex vivo model of the skin-implant interface of percutaneous OI implants. The model incorporated the complexities of the skin environment to reflect a clinical scenario and translate technologies toward in vivo models. We hypothesized that the oregano gel would attain a 4 log_10_ reduction in colony forming unit (CFU) counts within five days of gel administration.

## 2. Materials and Methods

### 2.1. Supplies, Instruments, and Reagents

Tryptic soy broth (TSB) was purchased from MilliporeSigma (Burlington, MA, USA), Petri dishes, agar, and general supplies/reagents from Fisher Scientific (Hampton, NH, USA), brain heart infusion (BHI) broth from Research Products International (Mt Prospect, IL, USA), and Baird Parker (BP) agar was from Remel Incorporated (San Diego, CA, USA). Cell lifters were purchased from GenFollower Biotech (Shaoxing, China). CDC biofilm reactors were purchased from Biosurface Technologies (Bozeman, MT, USA). Stomacher bags were purchased from Seward Laboratory Systems Inc (Bohemia, NY, USA). Multiple bottles of oregano essential oil were purchased from PlantTherapy (Murray, UT, USA). Oil compositions were obtained from the company and the main ingredients are located in Table 1:

### 2.2. Biofilm Growth

Biofilms of *S. aureus* ATCC 6538 were grown on Ti coupons; coupons were pre-treated for 24 h with hydrochloric acid (HCl; 70%) after which they were rinsed with DI water for 10 min and sonicated in DI water for 10 min to remove residual acid. The HCl roughened the surface for improved biofilm adherence. However, high concentrations of HCl etch the surface, removing titanium’s oxide layer [26]. To restore the oxide layer, the coupons were submerged in a 10:1 nitric acid:DI water solution for 30 min after which they were rinsed and sonicated with DI water for 10 min each (modified protocol from ASTM Standard B 600-91 [27]). Treated coupons were then placed in the holding arms of a CDC biofilm reactor, after which the entire system was autoclaved and placed on a hot plate at 37 °C and 130 rpm. The waste tube was then clamped off after which 500 mL of sterile 100% BHI was aseptically poured into the reactor. One mL of a 0.5 McFarland solution (using *S. aureus* that had been streaked on TSB and grown overnight) was then aseptically added to the reactor. Biofilms were grown in a 72 h cycle: 24 h 100% BHI batch phase, 24 h flow with 20% BHI (waste tube was unclamped), and 24 h flow with 10% BHI with a flow rate of 6.9 mL/min. After 72 h of growth, reactor arms were briefly immersed in sterile PBS to remove loosely adherent cells. Coupons were aseptically placed in sterile 6-well plates or Petri dishes until inoculation (biofilm was removed from the side of the coupon that faced the reactor wall).

### 2.3. Gel Formulation, Antibiofilm, and Zone of Inhibition (ZOI) Testing

The antibiofilm efficacy of oregano gel was unknown. We formulated and tested multiple iterations of an HA-based gel (1.75% *w*/*w*) and varied *w*/*w* concentrations of oregano (0.1%, 0.5%, 1%, 2%, and 10%) to determine the level of oregano required to attain good antimicrobial efficacy (defined by complete eradication of the biofilm grown on a Ti coupon). All gel components were added to a sterile syringe after which a syringe mixer was utilized to mix the gel back and forth ten times. The gel was then placed in a fridge (approximately 4 °C for 72 h) to promote the gelling properties of the HA. Biofilms grown on Ti coupons were submerged in 1 mL of gel and incubated at 37 °C for 24 h. Coupons were removed from the gel and gently dipped in PBS to remove residual gel. Coupons were then placed in 2 mL sterile PBS, vortexed for 1 min, sonicated (∼42 kHz) for 10 min, and vortexed again for 10 s. A 10-fold dilution series was implemented and 10 μL aliquots were plated on TSB agar. Plates were incubated at 37 °C for 24 h and colony forming units (CFU) were counted to determine the number of surviving organisms. CFU counts were then compared across the different gel concentrations to determine efficacy profiles of each.

Additionally, we performed zone of inhibition (ZOI) tests to confirm the antimicrobial potential of the oregano gel. The tests were also used to determine whether the HA base had inherent antimicrobial activity, which if present, could lead to misinterpretation of oregano outcomes. We tested two gel formulations: 2% oregano + 1.75% *w*/*w* HA and 1.75% *w*/*w* HA alone (no active antimicrobial ingredient). To perform ZOI analyses, a 0.5 McFarland standard of *S. aureus* ATCC 6538 was prepared. A sterile cotton swab was soaked in the solution, wrung out slightly, and streaked thoroughly over the surface of mannitol salt agar three times (each streaked lawn had a distinct direction). A 0.5 mL aliquot of gel was added to the middle of a plate and incubated at 37 °C for 24 h (*n* = 3 repeats per gel iteration). Plates were photographed and the ZOI measured.

### 2.4. Preparation of Ex Vivo Samples

Our ex vivo system was based on a translational in vivo OI implant ovine model (see Figure 1). We obtained the forelegs of sheep cadavers and shaved the skin below the proximal metacarpal joint. We removed the skin from the leg and cut the skin into approximately 1” × 1” squares. The skin was sterilized using a modified protocol by Huang et al.; skin was submerged in a static solution of 70% ethanol for 45 min followed by 10% bleach for an additional 45 min [28,29]. Three rinses of sterile DI water washed away residual bleach after which the skin was laid out flat between two pieces of sterile gauze and wrapped in sterile aluminum foil. Sterilized skin was preserved at −80 °C.

For the ex vivo tests, prepared sheep skin was thawed, trimmed to precisely 1” × 1” squares with a sterile scalpel, and the center of the skin was removed with a 6 mm biopsy punch. Using sterile surgical gloves, the skin pieces were aseptically pinned to sterile sponges (1/2” thick, ∼1.5” diameter) and a sterile three-inch Ti rod was pushed through the biopsied hole to simulate the OI implant. Additionally, modified 90 × 15 mm Petri dishes were used for structural support. They were sprayed with 70% isopropyl alcohol and placed around the sponges; a hole was removed from the bottom to allow for the skin (see Figure 2). During experimentation, the sponge was periodically filled with 20 mL of sterile DI water to minimize skin desiccation.

### 2.5. Skin Inoculation

The corner of a cell lifter was used to scrape biofilms off the Ti coupons. Specifically, the corner of the lifter was scraped down the center of a coupon (streak was ∼7 mm long) (see Figure 3). The collected biofilm was spread on the skin directly adjacent to the Ti post. Two scrapes of the coupon surface were taken, each being spread on half the circumference of skin surrounding the Ti post. Inoculated skin samples were covered with squares of sterile gauze (a slit through the center allowed the Ti rod to come through the gauze), secured with tape, and incubated in an upside-down position at 37 °C for 4 h; this provided a “conditioning” period for biofilms to begin adapting to the skin environment before antimicrobial treatment.

### 2.6. Ex Vivo Antibiofilm Testing

Oregano Group: After the conditioning period, 1.5 mL of 2% oregano gel was applied directly around the skin-implant interface (rationale for selection of 2% is provided in the Section 3); see Figure 4. The samples were then wrapped in sterile non-adhesive bandages (slits were made in the center of each bandage to allow the Ti rod to protrude through). Samples were placed in the incubator (37 °C) and every 24 h, fresh gel and bandages were re-applied. Twenty mL of sterile DI water was used to hydrate the sponge before samples were returned to the incubator. Samples were incubated for up to five consecutive days.

Positive Control (No Treatment) Group: Skin samples were inoculated with bacteria after which they were placed in the incubator without any treatment. Bandages were changed daily and sponges hydrated with 20 mL sterile DI water with each change. Samples were incubated for up to five consecutive days.

Number of Samples: For each day of treatment (in both oregano and positive control groups), 16 skin samples were evaluated for each group.

### 2.7. Microbiological Quantification

Ti Coupons: Coupons were dropped into conical tubes containing 2 mL PBS. They were then vortexed for 1 min, sonicated for 10 min, and vortexed for 10 s to knock the bacteria off the coupon surface and suspend it in the PBS. The bacteria was quantified using a ten-fold dilution series and plated on tryptic soy broth (TSB).

Cell Lifters: After cell lifters were scraped across the surface of the coupon, the cell lifter was cut with a sterile pair of scissors 1-inch from the end. The cut-off end (with biofilm) was aseptically dropped into a 10 × 100 mL glass test tube filled with 2 mL of PBS. Lifters were vortexed for 1 min, sonicated for 10 min, and vortexed for 10 s to suspend the bacteria from the lifter in the PBS. The bacteria was quantified using a ten-fold dilution series and plated on TSB.

Skin Samples: Pins and the sponges were removed and skin samples were gently rinsed for five seconds in PBS to remove residual gel before being weighed. Each sample was placed in a stomacher bag with 2 mL PBS and stomached using a Triple Mix Paddle Blender for two minutes to break up biofilm and suspend bacteria in PBS. Samples were then quantified using a ten-fold dilution series before being plated in duplicate on TSB and Baird Parker agar (*S. aureus* selective).

### 2.8. Preparation of Samples for SEM Imaging

Ti Coupons: Coupons were removed from the CDC biofilm reactor and placed in 90 × 15 mm Petri dishes filled with Modified Karnovsky’s solution to 2 mm below the brim, wherein they sat for 2 h. The solution was then removed and replaced with increasing concentrations of ethanol for 2 h each: 70%, 90%, and 100%. Samples were covered and the ethanol was allowed to completely evaporate after which samples were gold sputter coated and imaged using secondary electron imaging (SEI) on a JEOL JSM-6610 SEM.

Skin Samples: Samples were placed in disposable specimen containers (960 mL) filled 3/4 of the way with Modified Karnovsky’s solution. The solution was replaced once a day for three days. The skin was dehydrated in graded solutions of ethanol (three days at each grade, new ethanol each day): 70%, 80%, 90%, and 100%. On the final day, samples were covered and the ethanol was allowed to evaporate after which the skin was gold sputter coated and imaged by SEM using the SEI setting.

## 3. Results

### 3.1. Biofilm Growth

We observed that to achieve repeatable biofilm growth, the Ti coupons needed to be freshly etched with hydrochloric acid (HCl) approximately every six reactor cycles. Without this periodic HCl treatment, growth became inconsistent/patchy across the coupon surfaces. Additionally, coupons were thoroughly cleaned after each cycle with 10% bleach (removing the oxide layer) and passivated with nitric acid. Altogether, regular acid treatments resulted in consistent biofilm growth across all coupons (9.50 ± 0.36 log_10_ CFU/coupon) and cell lifters removed a repeatable amount of biofilm each time (6.00 ± 0.22 log_10_ CFU/skin sample).

### 3.2. Gel Formulation, Antibiofilm, and Zone of Inhibition (ZOI) Testing

After three days of chilling in the refrigerator, the gel reached a viscosity and consistency similar to honey. Oregano oil naturally separated out from the phosphate buffer solution (PBS) within 24 h and was remixed immediately before use to resuspend the oregano oil throughout the gel. Initial antibiofilm experiments to test the efficacy of the oregano oil gel iterations showed that biofilms were reduced to below detectable limits at concentrations ≥ 1% (see Figure 5). Based on the results of the initial antibiofilm testing, the 2% percent oregano gel was selected for multi-day experiments because it was a comparable concentration to other concurrent studies in our lab and 10% oregano had higher potential for toxicity; oregano oil is known to cause irritation [12].

ZOI tests provided an indication of the antimicrobial activity of the HA base and active gel components. Specifically, oregano-containing gels showed a clear zone of kill while the HA only gel did not demonstrate any antimicrobial activity (see Figure 6). The ZOI for oregano averaged 39.23 ± 3.89 mm in diameter (measured using ImageJ).

### 3.3. SEM Imaging

Scanning electron microscopy (SEM) imaging was used to determine biofilm community morphology (Figure 7a) and confirm that a single scrape of the cell lifters was sufficient to remove the basal regions of the biofilm communities from the surface of the Ti coupons used for skin inoculation; the goal was to apply all biofilm components, including extracellular polymeric substances (EPS), onto the surface of the ex vivo skin samples. Imaging indicated that cell lifters removed the entire depth of the biofilm, including the EPS, leaving the scraped area of the Ti coupon largely free of cells (see Figure 7b). Furthermore, skin images showed clumps of cells scattered across the skin surface (see Figure 8), indicating cells exhibiting the properties of biofilms were transferred to the skin; we anticipated that individual cells would have been observed if planktonic cells had been spread over the skin and not biofilms.

### 3.4. Ex Vivo Multi-Day Antibiofilm Testing

Quantification of the skin bacterial load was performed using both tryptic soy broth (TSB) and Baird-Parker agar. The rationale was based on observations during the multi-day experiment; although no contamination was observed after Day 1 of treatment, data from subsequent days of treatment revealed bacterial growth on the skin that conflated *S. aureus* results. This occurred despite the use of aseptic techniques while changing bandages and administering treatment. Thus, Baird-Parker agar (*S. aureus* selective) was utilized to resolve *S. aureus* counts while TSB assessed natural flora and/or exogenous contaminants that could have traversed the permeable bandages throughout the multi-day experiment.

Ex Vivo application of oregano gel in the multi-day study revealed a reduction of the biofilms (approximately 6–7 log_10_ by Day 4 of treatment (see Figure 9)). With the exception of Day 2, each day of gel application resulted in a statistically significant reduction in colony forming unit (CFU) counts compared to the positive control group (*p*< 0.05, student *t*-test). The 4 log_10_ reduction (unofficial benchmark recommended by the FDA for antimicrobial claims) was observed on Day 4 while statistically significant CFU reductions (*p*< 0.05, student *t*-test using Positive Control Group as comparator) were observed nearly every day of treatment (see Figure 9). Interestingly, an increase in CFU count occurred on Day 2 and Day 5 in comparison to the previous day of treatment.

We were unable to assess if the CFU count increase in the oregano groups (after Day 4) would continue to progress with additional days of gel application as the skin naturally deteriorated beyond use after five days.

## 4. Discussion

Patients with percutaneous osseointegrated (OI) implants are frequently affected by implant-related infections, resulting in persistent irritation, discomfort, and pain associated with their implants [30]. Even brief trips outdoors, with risk of biofilm exposure, can lead to challenging infections requiring antibiotic treatment. It is thus imperative to manage biofilm contamination at the skin-implant interface by developing effective technologies that target the biofilm phenotype. Better management of biofilm contamination may significantly decrease the prevalence of infection. The experiments presented herein provide an ex vivo model to support these developmental efforts.

The ZOI experimentation successfully demonstrated the HA-based gel without an active ingredient had no antimicrobial activity. As a result, we did not feel it was necessary or productive to spend resources and time on an ex vivo negative control group. The oregano gel ZOI’s had to be repeated multiple times due to an unexpected outcome. If the volume of oregano gel added to the MSA plate agar defect did not remain completely below the edge of the defect, the gel would melt in the incubator, dispersing the oregano across the entire surface of the agar. This produced inaccurate results as it increased the initial 6 mm starting zone, resulting in inhibited growth across the entirety of the agar.

The ex vivo model was designed to mimic a clinical paradigm as well as an in vivo ovine model on which these therapies will be tested in the future. More specifically, it modeled utilizing biofilms as initial inocula to represent a situation where biofilms may reside at the skin-implant interface of a patient’s OI implant. We confirmed mature biofilms were present on Ti coupons (as indicated by scanning electron microscopy (SEM) images) and the basal regions of the biofilm were collected and successfully inoculated onto the ex vivo skin samples (likewise indicated by SEM imaging). Each Ti coupon in the CDC biofilm reactor routinely developed ∼10^9^ CFU/coupon, which would be an unrealistic burden level at the skin-implant interface of an OI implant; if such a level were present, surgical intervention would be likely. We attempted to more closely model a real-life scenario where approximately 10^6^ colony forming units (CFU) would be present. To achieve this, we performed multiple biofilm collection methods (unpublished) to transfer a lesser yet repeatable amount of biofilm to the skin. As presented, we determined the corner of a cell lifter was the most repeatable and straightforward tool, while the cell lifters allowed consistent biofilm inoculation across samples, this process still resulted in biofilm levels upwards of 10^6^ CFU/g.

There is discussion in the biofilm community as to whether high numbers of bacterial inocula are appropriate in given systems. The “rule of thumb” is that 10^5^ CFU/g tissue cause infection, with infectious doses lowering in the presence of a biomaterial [31,32]. The described ex vivo model inoculum was well above the CFU count “rule of thumb” required to represent an infection and was potentially excessive. Yet, the rationale for inoculating with higher numbers of bacteria is two fold: (1) high number inocula are used to ensure a positive signal of infection develops in animal models of infection [33], and (2) the FDA (unofficially) requests to see a 4 log_10_ reduction in bacterial burden when testing or making claims about antimicrobial technologies. Because limits of detection are often below 10^2^ CFU in 10-fold dilution plating, a baseline inoculation of 10^6^ CFU is necessary to resolve a 4 log_10_ reduction. Thus, we selected an inoculation level slightly upwards of 10^6^ CFU/sample.

We also desired to inoculate the sites with mature biofilms. We considered maturity as constituting biofilm communities with three-dimensional structure, extracellular polymeric substances (EPS), and signs of water channel formation. We observed that cellular count within biofilm communities was essentially the same after both 48 and 72 h of growth, but extended growth time did result in a greater degree of EPS formation and community robustness–an observation consistent with previously published work [34]. Additionally, since biofilm density is known to influence antibiotic tolerance, we determined to grow biofilms for a total of 72 h to produce biofilms that may be challenging to manage [35].

Experimenting with the gel in a biofilm-inoculated ex vivo model, we collected multi-day data points since topical therapies are commonly administered for multiple days in clinical settings. We initially planned to collect 14 days of data and began experiments using skin samples pinned to a silicone platform. Within 2 h of those initial experiments, the skin effectively became a dried chip, resulting in biofilm death independent of gel treatment. We adjusted the setup and used sponges hydrated with sterile DI water to maintain moisture, which improved data collection. Unfortunately, despite rehydration, by Day 5 the ex vivo sheep skin samples began to lose consistency and degrade—also challenging data collection. A physiological buffer solution (PBS) was tried as an alternative to DI water, but it quickly evaporated and left precipitate on the surface of the skin, which had the potential to alter the skin surface on which the biofilm resided and produce misleading results. A nutrient broth solution was also considered to rehydrate samples. Nevertheless, the ex vivo samples were placed in an upside down orientation (surface of skin facing down) to simulate the effects that gravity may have had on the gel administration as transfemoral OI implants have similar orientation. As a result, the nutrient broth would need to be fed into the system through the back of the sponges, infusing both the back and surface of the skin (site of biofilm inoculation) with broth. Thus, while effectively hydrating the skin, the broth would also feed the biofilms and skew data collection. Considering these paradigms, multi-day experiments were shortened from 14 to five days. Further, the skin degradation we observed may have contributed to the slight increase in biofilm burden in the oregano skin samples after Day 4; degrading tissue could have provided additional nutrients and/or physical regions for bacteria to penetrate, and into which gel products failed to diffuse.

The need for both Baird-Parker and tryptic soy broth (TSB) for accurate quantification may have seemed excessive. It is possible contamination could have resulted from a permeable bandage or bacteria deep in skin pores the sterilization process did not eradicate. Those two issues could have been addressed by keeping the samples in an aseptic environment throughout the study or a more thorough sterilization process. Nevertheless, in an in vivo model, contamination is unavoidable, thus we chose to maintain the environment to more closely model an in vivo scenario; allowing natural contamination ex vivo helped determine an accurate method of bacterial quantification for future in vivo studies.

While 2% oregano gel demonstrated an overall trend of decreasing bioburden over multiple days of application, there was an interesting spike in CFU count on Day 2 of treatment that was not observed in the positive control group. This trend may be explained two-fold: potential triggering of the SOS pathway may have caused a brief increase in bacterial growth in the oregano group [24] or the lack of aqueous gel in the positive control group may have modified the exposure environment (a limitation of our experimental set-up).

Despite this spike, 2% oregano gel administration overall demonstrated promising antibiofilm potential when used over multiple days. Other studies have shown greater bacterial reduction in a shorter amount of time [22,23,24,25,36], but utilized planktonic bacteria for testing. As most data collected with oregano involves planktonic bacteria, the results of our experiments are difficult to compare to other existing studies except on a broad scale. The 2% oregano gel had a higher concentration than what exists in the majority of antimicrobial studies as higher oregano concentrations are typically used in alternative situations such as improving wound healing [22,25]. Furthermore, we used high density biofilms as initial inocula. Oregano efficacy is oft reported using minimum inhibitory concentration (MIC) results against 10^5^ suspensions of planktonic bacteria [37], which was far less than the 10^6^ CFU count from our biofilms. Our use of robust biofilms and their naturally higher cell count likely necessitated the greater percentage of oregano. Additionally, studies incorporating biofilms into their experimental procedures have demonstrated oregano oil can inhibit biofilm formation, but did not explore oregano oil for eradicating established biofilms [12,13]. These considerations support the need for targeted biofilm analysis as biofilms display greater recalcitrance to antimicrobial therapy compared to planktonic bacterial outcomes.

The 2% oregano concentration was lower than the up-to-50% concentrations recommended for at-home use, while biocompatibility or cytotoxicity studies have not been performed using high concentrations (e.g., 50%). We consider that higher amounts of oregano oil may be unnecessary and lead to skin irritation or other complications without added benefit. However, while we cannot make clinical recommendation, we suggest there is an important balance between cytotoxicity and efficacy (recognizing that infection itself is cytotoxic, which, in clinical situations, would be an important consideration).

Altogether, oregano as a proof-of-concept gel displayed promise as a biofilm management therapy for the skin-implant interface of OI implants, but there remain points to consider in moving forward. For example, the oregano oil naturally separated in PBS solution, necessitating aseptic mixing before each use, which would add complexity to patient application of the gel. Emulsifiers were originally put into the gel recipe, but we determined that common emulsifiers, such as DMSO [12,38] had antimicrobial activity against our strain of *S. aureus* (unpublished) that would influence our analysis of oregano’s antimicrobial activity. In the future, we will consider a nanoemulsion similar to Taleb et al., or formulate a new gel incorporating a non-antimicrobial emulsifier [12].

Further, results supported our hypothesis concerning the 4 log_10_ CFU reduction benchmark for oregano oil formulated into a 2% *w*/*w* gel, but observations were limited by the five day skin usage. Further validation is needed with an in vivo model to more fully elucidate their efficacy and determine if the log_10_ reduction can be maintained beyond five days of gel administration.

## 5. Conclusions

In conclusion, despite limitations in our ex vivo experiments, the results of our study demonstrated an oregano oil gel’s potential as a biofilm management treatment. In Vivo studies are the next step to further understand its ability to prevent infection at the skin-implant interface of percutaneous OI implants.

## Figures and Tables

**Figure 1 microorganisms-10-02133-f001:**
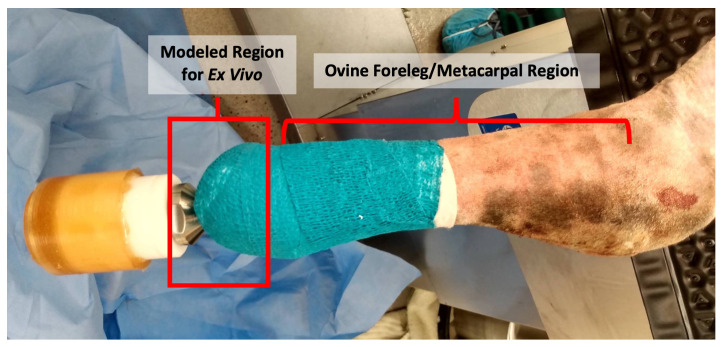
In Vivo ovine model immediately post surgery (from a separate IACUC-approved study). The ex vivo model was developed to simulate this situation, using sheep skin obtained from the metacarpal region of sheep cadavers and maintained in an upside down position to represent the sheep standing upright.

**Figure 2 microorganisms-10-02133-f002:**
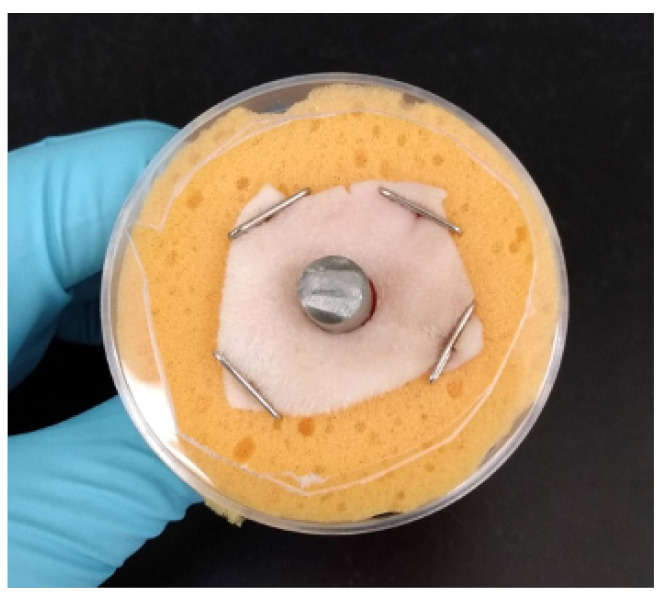
Example of assembled ex vivo model with Petri dish for structural support.

**Figure 3 microorganisms-10-02133-f003:**
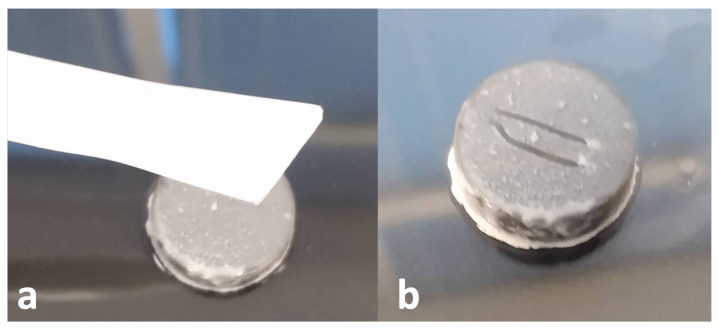
(**a**) Cell lifter next to Ti coupon with 72 h of *S. aureus* biofilm growth (**b**) Ti coupon after two 7 mm long streaks of biofilm have been removed from the surface. The lines avoid the edges of the coupon where the most inconsistent/variable biofilm growth occurs.

**Figure 4 microorganisms-10-02133-f004:**
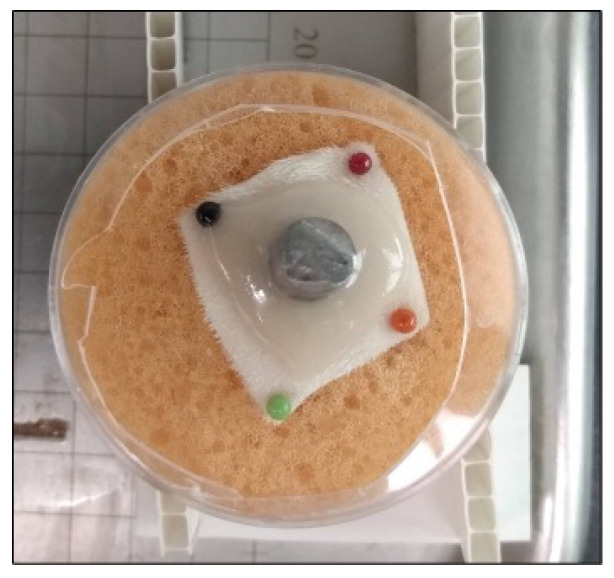
Ex Vivo skin sample with 1.5 mL oregano gel.

**Figure 5 microorganisms-10-02133-f005:**
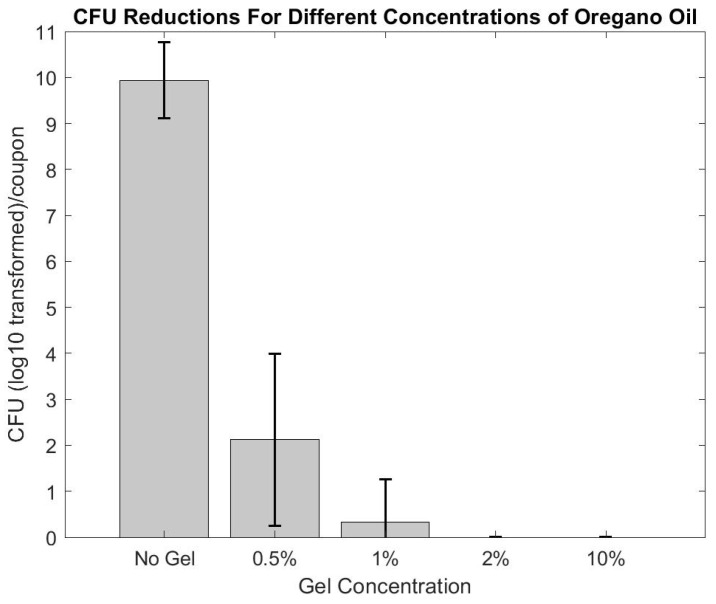
Various concentrations of oregano oil were tested to determine the ideal amount of oregano to use in the multi-day studies. 2% oregano and upwards caused complete eradication of the biofilm (within detectable limits).

**Figure 6 microorganisms-10-02133-f006:**
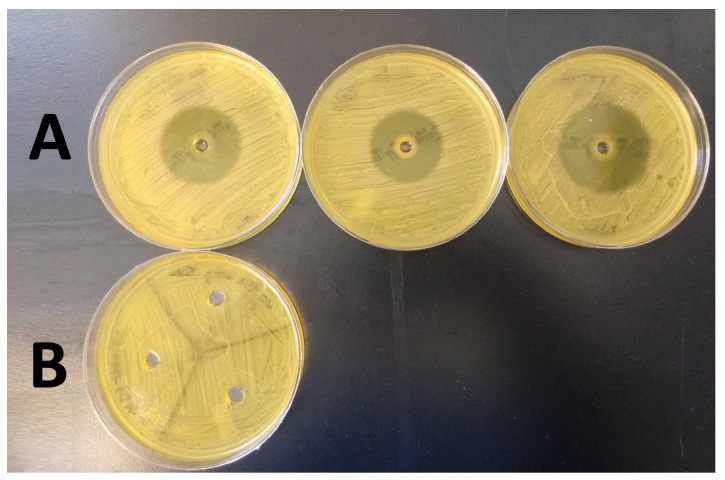
Results of the ZOI tests performed in triplicate for 2% oregano oil gel (Row (**A**)), and HA only gel (Row (**B**)).

**Figure 7 microorganisms-10-02133-f007:**
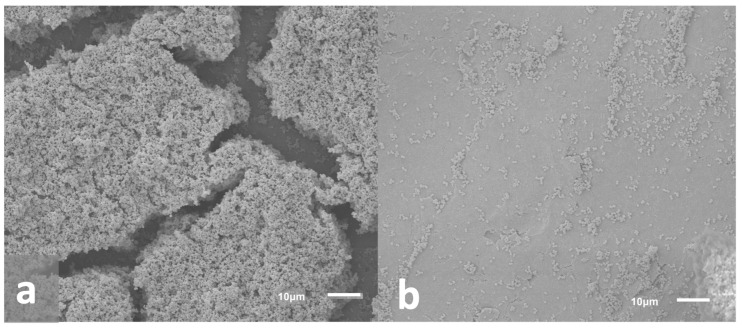
Representative SEM images of biofilms on the surface of Ti coupons. (**a**) Biofilm growth on Ti coupon prior to removal. Images indicated that biofilms grew to maturity with three-dimensional structures and likely water channels (caverns between structures). (**b**) Biofilm remaining on Ti coupon after removal with a cell lifter. Images indicated that cell lifters removed the majority of biofilm components including the basal layer.

**Figure 8 microorganisms-10-02133-f008:**
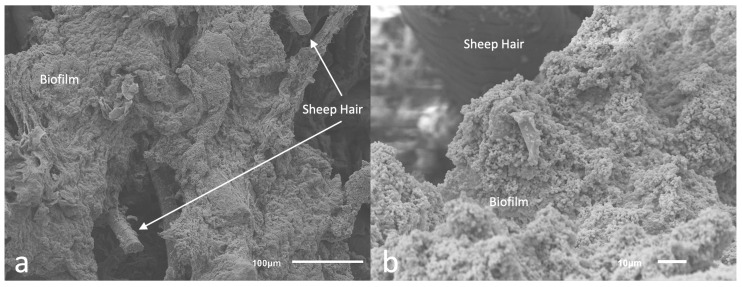
Skin inoculated with biofilm. (**a**): Zoomed out image where sheep hair can be observed on the surface of the skin inoculated with biofilms (**b**) Closer look at the cells composing the biofilm inoculated onto the sheep skin.

**Figure 9 microorganisms-10-02133-f009:**
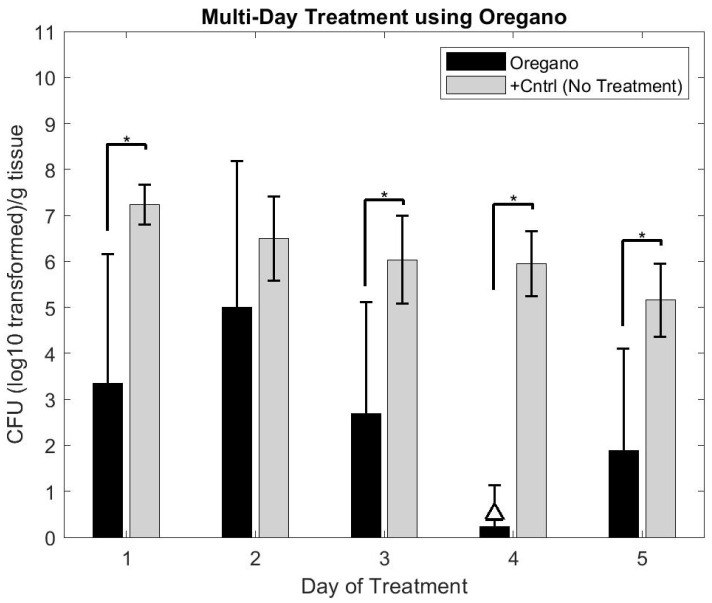
Graphical representation of biofilm burden on skin after multiple days of treatment with oregano gel in comparison to positive control samples, which received no treatment. The triangle (△) indicates the treatment that attained a 4 log_10_ reduction in CFU counts compared to that day’s positive control CFU count. Oregano gel only attained this level of reduction on Day 4. However, statistically significant results (student *t*-test compared to positive control, *p* < 0.05) were observed on all but one day of treatment. * Indicates *p* < 0.05.

**Table 1 microorganisms-10-02133-t001:** Three main components of the oregano essential oil utilized in this study. Percentages were obtained from data sheets available on the company’s website (https://www.planttherapy.com/, accessed on 4 August 2022).

Component	% Present
Carvacrol	69.4–76%
β-Caryophyllene	1.1–2.3%
Thymol	3.7–3.8%

## Data Availability

Not applicable.

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
