# Peer review of "Determining the Antibiofilm Efficacy of Oregano Gel in an Ex Vivo Model of Percutaneous Osseointegrated Implants"

_microorganisms, 2022, doi:10.3390/microorganisms10112133_

Round 1
Reviewer 1 Report
Interesting article on the efficacy of oregano oil on disrupting biofilm
Minor changes:
Line 49 in the literature
Methods:
% or molarity of HCl used
Line 82 with a flow rate of
Line 142 – repeated words in sentence
References:
Please check your references to ensure that they are complete eg Ref 17 to 21 are incomplete
Please use italics for bacterial names eg Ref 33 Staphylococcus pyogenes and Ref 38 Staphylococcus aureus
Author Response
Dear Reviewers and Journal Editors,
Thank you for considering our publication and the feedback provided. The comments and edits are well-received. Our responses to each reviewer are below:
Reviewer 1:
- Interesting article on the efficacy of oregano oil on disrupting biofilm
- Minor changes:
- Line 49 in the literature
- Methods:
- % or molarity of HCl used
- Line 82 with a flow rate of
- Line 142 – repeated words in sentence
- References:
- Please check your references to ensure that they are complete eg Ref 17 to 21 are incomplete
- Please use italics for bacterial names eg Ref 33 Staphylococcus pyogenesand Ref 38 Staphylococcus aureus
- Thank you for your review and comments for improving format, references, and wording. Each of the changes have been made and are shown with Track Changes in the updated manuscript.
Reviewer 2 Report
1. Why ZOI has been performed on TSA agar, not MHA plates?
2. Line 142: The samples were then samples were - should be corrected.
3. Why no negative control was included in ex-vivo?
4. Fig. 6 - plates are not properly inoculated.
5. Line 223 - have you identified what bacteria grown on the plates? How did you confirm that on selective agar solely your S. aureus was grown, and not the other S.aureus (contamination).
6. Fig. 9 - error bars are huge. it is suspicious that it is statistially significant decrease. How would you explain the increase on day 2 and day 2 in oregano group? why there is no such effect in control group?
7. what was the ZOI results for other concentrations? Did you try any control with strain resistant to oregano? Why 0.5 mL was applied? It is big amount of gel.
8. Line 253, 254 - information about gaseous state sound speculative and should be removed if you have no scientific proof for that.
Overall, I can not find any scientific merit and value in this research. I don't think this may be applied in clinics. Also the novelty of the research is questionable - properties of oregano oil are know since decades, a lot of similar studies have been already performed and this oil is not approved and commonly used as alternative to antibiotics. I do not see any benefits to use it instead of standar therapies. Also the ex vivo model - I do not understad why ovine model has been chosen, if it is possible to use human skin to perform such test. Methods used to count biofilm seems not be appropriate - please, refer to novel ublications in this topic. Neccessary positive and negative controls should be used in each case. A lot of citations are older than 10 years, it's long time in science.
Could you please indicate the Ethics Commitee consent is neccessary or not - and in not, then: why?
Author Response
Reviewer 2:
- Why ZOI has been performed on TSA agar, not MHA plates?
- We have modified our ZOI and performed the tests on MHA plates
- Line 142: The samples were then samples were - should be corrected.
- This was corrected.
- Why was no negative control was included in ex-vivo?
- The gel base alone was screened in vitro to determine its potential antimicrobial effect. Due to its lack of activity in the simple screen wherein antimicrobial activity would have been much easier to achieve, we did not feel it was necessary or productive to spend resources and time on negative control work ex vivo.
- 6 - plates are not properly inoculated.
- Plates were redone with proper inoculation
- Line 223 - have you identified what bacteria grown on the plates? How did you confirm that on selective agar solely your aureus was grown, and not the other S.aureus (contamination).
- Great question. When grown on TSB, the contaminant colonies were not indicative of aureus or any staphylococcal species. Nevertheless, to confirm only our strain of bacteria (S. aureus ATCC 6538) grew on the selective agar, we grew it from frozen stocks and recorded colony morphology, color, smell, and appearance. Comparison of the colony appearances between the frozen stocks and the skin samples were similar in all aspects. Definitive determination would require PFGE, but we feel for this ex vivo analysis, it is beyond the scope of necessity.
- 9 - error bars are huge. it is suspicious that it is statistically significant decrease. How would you explain the increase on day 2 and day 2 in oregano group? why there is no such effect in control group?
- We agree, the error bars are broad. But we also wanted to present raw data as opposed to biasing outcomes by rerunning experiments multiple times to observe the data we “wanted.” We present the outcomes as is. The variability may reflect one of the central problems with essential oil applications: variability can arise as these are not standardized or strictly defined materials, i.e., a leaf of one extract may have more active agent than another. Yet we experimented with the same bottle of oregano oil so this issue was less of a risk.
- As total bioburden is presented, the increase in CFU on Day 2 is an interesting phenomenon. One potential explanation may be due to the broad antimicrobial mechanisms of oregano oil. Cui et al. 2019, states the majority of activity results from irreversible damage to the cell membrane, but DNA processes are also affected, potentially triggering the SOS pathway (bacterial defense mechanism). As DNA damage seems to be a minor antimicrobial mechanism, the SOS pathway may prevent bacterial death for a short time period until the bacteria succumbs to the main antimicrobial mechanism (cell membrane damage) for which it does not have a defense mechanism. This remains speculation as we were unable to find multi-day oregano exposure data in the literature limiting comparative outcomes. Nevertheless, the overall trend of observing a decrease in bioburden over time motivates further testing.
- The lack of increase in the controls may be due to the lack of an aqueous gel, which modifies exposure environment and may increase growth of some species. We consider determining the exact parameters beyond the scope of the focused study. However, we have noted this as a limitation in the Discussion.
- What was the ZOI results for other concentrations? Did you try any control with strain resistant to oregano? Why 0.5 mL was applied? It is big amount of gel.
- The overall purpose for performing ZOI analysis was to ensure that in comparison, the HA only gel did not have antimicrobial effect compared to the selected 2% w/w oregano gel, thus other concentrations were not tested.
- We do not have a strain resistant to oregano, but do not feel that would be necessary for this proof-of-concept phase of testing.
- We modified the ZOI to utilize a volume of gel consistent with an agar defect the size of a Kirby-Bauer disk (approximately .08 mL)
- Line 253, 254 - information about gaseous state sound speculative and should be removed if you have no scientific proof for that.
- ZOI’s were redone and new findings have removed the need for this speculation
- Overall, I cannot find any scientific merit and value in this research. I don't think this may be applied in clinics. Also, the novelty of the research is questionable - properties of oregano oil have been known for decades, a lot of similar studies have been already performed and this oil is not approved and commonly used as alternative to antibiotics.
- We invite the reviewer to reread the Introduction to consider the scientific value. There are OI patients literally putting oregano products on their skin with no rationale as to concentration consideration or experimental outcomes. There are no studies assessing the effect of oregano on the skin-implant interface of OI implants. Thus, the scientific merit is there.
- OTC essential oil products are used in many clinics. We also believe there are far too many used without scientific support, which underpins our interest in this study. Yet we do not agree that this wouldn’t be considered if experimentally supported.
- Yes, there is important rationale for why essential oils are used as alternatives, e.g., the problem of antibiotic resistance. Which further motivates this study.
- I do not see any benefits to use it instead of standard therapies. Also the ex vivo model - I do not understand why ovine model has been chosen, if it is possible to use human skin to perform such test.
- We are also examining standard therapies, but those will be published separately as the work is based on a pipeline of testing. Yet the rationale for testing oregano is provided above and in the Introduction.
- Ovine skin was chosen as this testing will be translated to an in vivo ovine model. We wanted the foundational testing to be relevant to the animal model we will perform.
- Methods used to count biofilm seems not be appropriate - please, refer to novel publications in this topic.
- We disagree. Vortexing, sonication, and 10-fold dilution plating continue to be the most common and validated biofilm quantification methods.
- Necessary positive and negative controls should be used in each case.
- The gel base alone was screened in vitro to determine its potential antimicrobial effect. Due to its lack of activity in the simple screen wherein antimicrobial activity would have been much easier to achieve, we did not feel it was necessary or productive to spend resources and time on negative control work ex vivo.
- A lot of citations are older than 10 years, it's long time in science.
- Older citations concerning prevalence of infection have been updated with newer articles with more accurate reflections of the issue
- There are a few reasons for the older citations as follows:
- A number of citations are used to reference the properties of biofilms which have been known for years and have not changed. Instead of citing newer articles with the information, we chose to use the original articles
- Oregano oil as an antimicrobial agent against bacterial skin inoculums does not appear to be a popular research topic based on literature review. As a result, there are few relevant articles to reference in comparing oregano against planktonic bacteria or biofilms within the last 10 years. A literature search revealed that recent oregano-related papers are focused on new methods of encapsulating or delivering oregano oil, or a reassessment of MIC values.
- Could you please indicate the Ethics Committee consent is necessary or not - and in not, then: why?
- Ex vivo testing does not require IACUC or ACURO approvals. These are only required for live animal testing.
Reviewer 3 Report
Very interesting study, all used methods are accurate, well described. In vitro results are promising.
Author Response
Reviewer 3:
- Very interesting study, all used methods are accurate, well described. In vitro results are promising.
Thank you for the feedback
Reviewer 4 Report
Composition of essential oils can change according to climatic, agrotechnical and other factors. How would the authors standardize the composition of their gel? Please give the composition of the used oregano oil. What can be the reason for CFU count increase in multi day treatment?
Author Response
Reviewer 4:
- How would the authors standardize the composition of their gel? Please give the composition of the used oregano oil.
- The main components of gel composition have been added to the paper. One objective of this paper was to resemble a real-life scenario where OI patients are purchasing off the shelf products and applying it to their limbs. As a proof-of-concept study, standardization of oregano is beyond the scope of the project. Nevertheless, we recognize that if a product (with claims) were to be marketed, standardization would be required, which raises an important question as oregano oil composition will vary based on plant origin, time of harvesting, exact distillation process. If this advances toward clinical application, an important step would be to assess the buffer zone of “2% oregano”. If oregano oil obtained from any supplier during any time of year and from any country of origin, produces the same antimicrobial activity at a 2% concentration, no modification to the gel composition is necessary. In the case that significant differences are observed, a single supplier may need to be utilized, and oregano composition definitively defined.
- What can be the reason for CFU count increase in multi day treatment?
- As total bioburden is presented, the increase in CFU on Day 2 is an interesting phenomenon. One potential explanation may be due to the broad antimicrobial mechanisms of oregano oil. Cui et al. 2019, states the majority of activity results from irreversible damage to the cell membrane, but DNA processes are also affected, potentially triggering the SOS pathway (bacterial defense mechanism). As DNA damage seems to be a minor antimicrobial mechanism, the SOS pathway may prevent bacterial death for a short time period until the bacteria succumbs to the main antimicrobial mechanism (cell membrane damage) for which it does not have a defense mechanism. This remains speculation as we were unable to find multi-day oregano exposure data in the literature limiting comparative outcomes. Nevertheless, the overall trend of observing a decrease in bioburden over time motivates further testing.
Reviewer 5 Report
The paper by Ong et al describes an original ex vivo model of percutaneous osseo-integrated implants, by which antibiofilm activity of an oregano gel was assessed. A mature biofilm of S.aureus was used as contaminant and a 4 days long observation was pursued. The working hypothesis has been correctly formulated and the paper is well written. Yet, the microbiological data relative to the efficacy of such oregano-containing gel in their ex vivo model are limited to the use of a single condition, also the control used as negative counterpart should be clarified. The titles and captions of the figures do not reflect the style of the journal. Please, modify.
Specific points
- Lanes 69-75: to facilitate reading and understanding, the procedure detailed in such paragraph should be clearly mentioned as pretreatment of the Titanium coupons
- Lanes 102-104 & Figure 6: the assessment of the HA (the only ingredient of the gel??) should be better described (doses…)
- The ZOI assay uses 0.5 ml aliquots of solutions to be tested; why such a large volume?
- Lanes 152-153: please clarify the numbers of the sample tested (16 totally tested samples; or 16 ctrl plus 16 gel-treated samples?)
- The Y –axis name of figures 5 and 6 should be revised (the term “transform” is unclear)
- Lane 221: “Skin quantification” should rather be “Quantification of the skin contamination…”
- Figure 9: in the text, the authors mention a comparison between each result and that of the day before; in the figure, the statistical analysis appears between ctrl and treated samples, within the same day; please clarify. In the figure title, the acronym CZ-01179 is indicated, what does it mean?
Author Response
Reviewer 5:
- Lanes 69-75: to facilitate reading and understanding, the procedure detailed in such paragraph should be clearly mentioned as pretreatment of the Titanium coupons
- Thank you. This has been modified.
- Lanes 102-104 & Figure 6: the assessment of the HA (the only ingredient of the gel??) should be better described (doses…)
- Description edited
- The ZOI assay uses 0.5 ml aliquots of solutions to be tested; why such a large volume?
- We modified the ZOI to utilize a volume of gel consistent with an agar defect the size of a Kirby-Bauer disk (approximately .08 mL)
- Lanes 152-153: please clarify the numbers of the sample tested (16 totally tested samples; or 16 ctrl plus 16 gel-treated samples?)
- Clarified
- The Y –axis name of figures 5 and 6 should be revised (the term “transform” is unclear)
- This was modified
- Lane 221: “Skin quantification” should rather be “Quantification of the skin contamination…”
- This was modified.
- Figure 9: in the text, the authors mention a comparison between each result and that of the day before; in the figure, the statistical analysis appears between ctrl and treated samples, within the same day; please clarify. In the figure title, the acronym CZ-01179 is indicated, what does it mean?
- This was modified
- Yet, the microbiological data relative to the efficacy of such oregano-containing gel in their ex vivomodel are limited to the use of a single condition
- This is true, but this single condition may be applied to many other scenarios that may experience day-to-day biofilm contamination, such as catheter ports and bandages for slow-healing wounds.
- The control used as negative counterpart should be clarified
- We did not use a negative control. The gel base alone was screened in vitro to determine its potential antimicrobial effect. Due to its lack of activity in the simple screen wherein antimicrobial activity would have been much easier to achieve, we did not feel it was necessary or productive to spend resources and time on negative control work ex vivo.
- The ZOI assay uses 0.5 ml aliquots of solutions to be tested; why such a large volume?
- We modified the ZOI to utilize a volume of gel consistent with an agar defect the size of a Kirby-Bauer disk (approximately .08 mL)
Round 2
Reviewer 5 Report
The paper has been significantly improved